# Clients' perspectives on the utilization of reproductive, maternal, neonatal, and child health services in primary health centers during COVID-19 pandemic in 10 States of Nigeria: A cross-sectional study

**Babatunde Adelekan**[1], **Erika Goldson**[1], **Lorretta Favour C. Ntoimo**[2,3], **Osaretin Adonri**[1], **Yakubu Aliyu**[1], **Matthew Onoja**[1], **Idowu Araoyinbo**[1], **Emilene Anakhuekha**[1], **Ulla Mueller**[1], **Eno-Obong Ekwere**[4], **Micheal Inedu**[4], **Olayinka Moruf**[5], **George Swomen**[5], **Brian Igboin**[3], **Friday E. Okonofua**[3,6,7]*

1 United Nations Population Fund, Abuja, Nigeria, 2 Department of Demography and Social Statistics, Federal University Oye-Ekiti, Oye-Ekiti, Nigeria, 3 Women's Health and Action Research Centre, Benin City, Nigeria, 4 Education as a Vaccine, Abuja, Nigeria, 5 Planned Parenthood Federation of Nigeria, Abuja, Nigeria, 6 Centre of Excellence in Reproductive Health Innovation (CERHI), University of Benin, Benin City, Nigeria, 7 Department of Obstetrics and Gynecology, University of Benin, Benin City, Nigeria

* feokonofua@yahoo.co.uk

## Abstract

### Background

Reports from various parts of the world suggest that the COVID-19 pandemic may have severe adverse effects on the delivery and uptake of reproductive health, maternal, neonatal, and child health (RMNCH) services. The objective of the study was to explore women's experiences with utilization of RMNCH services during the COVID-19 pandemic in Nigeria, and to elicit their perceptions on ways to sustain effective service delivery during the pandemic.

### Methods

A cross-sectional survey of 2930 women using primary health care facilities for antenatal, delivery, postnatal, and child care services before and after the onset of the pandemic in 10 States of Nigeria were interviewed with a semi-structured questionnaire. Data were collected on women's socio-demographic characteristics and pregnancy histories, the services they sought before and after the pandemic, the challenges they faced in accessing the services, their use of alternative sources of health care, and their recommendations on ways to sustain RMNCH service delivery during the pandemic. The data were analyzed with descriptive statistics, and multivariable logistic regression using SPSS 20.0. All the statistical analyses were two-tailed with a 95% confidence interval, and the p-value was set at 0.05.

**Data Availability Statement:** The data underlying the results presented in the study are available from Zenodo 10.5281/zenodo.6128442.

**Funding:** The UN Nigeria Basket Fund in support of COVID-19 funded this study through the United Nations Population Fund (UNFPA) under the Civil Society Organisation Engagement (CSOE) project to reverse the negative impact of COVID-19 and ensuring access to essential services. The content is solely the responsibility of the authors and does not necessarily represent the official views of the funding organization.

**Competing interests:** The authors have declared that no competing interests exist.

## Results

The logistic regression results showed that women were at least 56% more likely to report that they used family planning, antenatal, and delivery services before the pandemic than after the pandemic started, but 38% less likely to report use of postnatal services. The experience of difficulty accessing RMNCH services was 23% more likely after the pandemic started than before the pandemic. Three categories of recommendations made by the respondents on measures to sustain RMNCH delivery during the pandemic included 1) facility improvement, and staff recruitment and re-training; 2) free and readily accessible PHC services, and 3) the provision of social safety nets including transportation and palliatives.

## Conclusion

We conclude that the COVID-19 pandemic limited women's access to antenatal, delivery, and childcare services offered in PHCs in Nigeria. Addressing the recommendations and the concerns raised by women will help to sustain the delivery of RMNCH services during the COVID-19 pandemic and future epidemics or health emergencies in Nigeria.

## Introduction

Reports from various parts of the world suggest that the coronavirus (COVID-19) pandemic may have severe adverse effects on the delivery and uptake of reproductive, maternal, neonatal, and child health (RMNCH) services [1–3]. This challenge has been attributed to the lockdowns associated with the pandemic and the diversion of material and human resources to the management of the virus and the provision of other emergency medical services [4]. While RMNCH services are also critically important and account for a substantial burden of disease and mortality especially in low-income countries [5–7], it is worrying that health systems may be unable to maintain a high level of convergence to enable the continuation of these services in the wake of the pandemic.

Our previous report interviewing primary healthcare workers in Nigeria [8] indicate that while Primary Health Centres (PHCs) across the country attempted to open their facilities for delivery of RMNCH services during and after the COVID-19 pandemic lockdowns, they faced several challenges in the delivery of services, including stock-outs and the difficulties in getting to service delivery points. A major challenge was the low utilization of the services by clients with a severe reduction in the number of women using the facilities as compared to the period before the pandemic. Even after the lockdown period, the number of women using the facilities has been slow to return to normal raising the question as to the experiences of women during the pandemic, and the alternative ways they sought the prevention and management of RMNCH challenges [8].

Nigeria has some of the most daunting indicators of reproductive, maternal, neonatal and child health in the world, and ranks among the countries with the lowest proclivity to achieve the health-related sustainable development goals [9, 10]. Poor access to evidence-based services offered in health facilities has been cited as a leading cause of the high rate of maternal, neonatal, and child mortality in the country [11–14]. The most recent evidence shows that 67% of women of reproductive age in Nigeria utilized skilled anatenatal care, 39% gave birth in a health facility, 42% received postnatal check in the first two days after birth, and use of modern family planning methods by currently married women is only 12% [10]. Given this notable challenge, it is important that Nigeria continues to identify various avenues to sustain the

delivery and uptake of RMNCH services despite the pandemic in order not to further compromise these indicators.

It is within this context that this study was conducted to investigate women's experiences with RMNCH services offered in PHCs before and after the COVID-19 pandemic started. The objective of this study was to investigate the COVID-19 experiences of women using identified RMNCH services before and after the onset of the pandemic, their experiences of various RMNCH challenges, and their health-seeking behaviour for these challenges. We hypothesize that women would have a limited propensity to use RMNCH services during the pandemic despite their continued experiences of reproductive and maternal health issues warranting the use of such services. We believe that the results would be useful in identifying innovative ways to improve women's uptake and demand for these services, and ensure the continuation of RMNCH services for the prevention of maternal and child morbidity and mortality during the pandemic.

## Materials and methods

### Study site

This study was conducted in 320 communities in 32 Local Government Areas (LGAs) across the Federal Capital Territory (FCT), Abuja, and 9 out of the 36 Nigerian States. The states were Lagos, Akwa Ibom, Kano, Kaduna, Gombe, Borno, Ogun, Enugu, Adamawa, and the FCT. The States were purposefully selected based on documented high rates of COVID-19 in the initial phase of the pandemic in Nigeria. Ten communities were selected from each LGA for the study, making 320 communities (villages) in all.

### Study design

This was a cross-sectional descriptive study. The study was initiated and coordinated by UNFPA Nigeria under the One UN Basket fund to respond to COVID-19 and implemented under the supervision of three national NGOs: The Women's Health and Action Research Centre (WHARC), Education as a Vaccine (EVA), and the Planned Parenthood Federation of Nigeria (PPFN). The three Implementing Partners (IPs) worked in partnership with three identified Civil Society Organizations (CSOs) per state to conduct the study. Overall, 30 CSOs across nine states and the FCT worked to conduct the study.

### Study population

The PHCs located in the communities were targeted and requested to identify women from their immediate communities who frequently use their facilities for RMNCH services. The inclusion criteria were: 1) women who registered as having received clinical services in the clinics per state before and after the pandemic started; 2) women receiving RMNCH and family planning services in the clinics; and 3) those who agreed to participate in the fully explained study.

The exclusion criteria were: 1) women who recently registered in the clinics without a pre-COVID-19 experience in the clinics. They were excluded because they were unlikely to be able to make comparisons with the pre-pandemic period; 2) women receiving other services apart from RMNCH and family planning services, and 3) women who refused to participate in the fully explained study.

### Sample size

Using the Yamane sample size formula [15], the estimated sample size per State was 384 (3,840 for nine States and FCT), with 10% adjustment for non-response, the total sample size was 422

per State or location (4220 for the nine States and FCT). About 13 women were to be interviewed in each of the 32 PHCs per State. 2930 women were successfully interviewed in 307 PHCs giving a non-response rate of 23.7%.

## Sampling procedure

The respondents were identified through the head of each PHC using a snowball approach in the catchment communities. The women were approached and interviewed either in their homes or when they presented for care in the PHCs. The recruitment continued until the sample size for each PHC/community is achieved.

## Data collection

The data were collected from November 1 to December 16, 2020 through an interviewer-administered questionnaire, which was programmed into Open Data Kit (ODK) on smartphones for interviewer-administered computer-assisted personal interviewing. The study protocol (questionnaire) was developed in WHARC, revised, and finalized by all IPs. Thereafter, each CSO identified the interviewers (data collectors) for health facilities in each LGAs. The data collectors were trained on quantitative data collection methods. After the training, the questionnaire was pretested in selected communities in each State.

The questionnaire consisted of a mix of structured and non-structured (open-ended questions) to answer the full range of the research questions. The questionnaire contained basic questions on the socio-demographic characteristics of the respondents, their knowledge, experiences, and practices of measures related to COVID-19, their experiences of utilization of RMNCH services in PHCs before the pandemic started, during the pandemic lockdown, and after the lockdown. The challenges of accessing care before, during the COVID-19 lockdowns, and after the lockdown were elicited. Also elicited were their perceptions and recommendations on ways to manage and improve services during and after the pandemic lockdowns. The COVID-19 lockdown took place in Nigeria in mid-March 2020 and was eased in September 2020. Thus, the period before the lockdown was identified as any time before March 15, 2020, while the lockdown period was between March 15 to the end of September 2020. The period after September 2020 when no lock-down occurred, and all schools, markets, and Churches were re-opened to users were defined as the post lockdown period. In this analysis, two periods are analysed–before the COVID-19 pandemic referring to any time before the lockdown, and after the pandemic started referring to the lockdown and post-lockdown periods.

## Variables and measures

Utilization of RMNCH services and deifficulty accessing the services were the outcomes variables. Utilization of RMNCH services were for family planning, antenatal care, delivery care, postnatal care, and child health (including immunization). Each outcome was coded one if accessed and received, and zero if not accessed and received. The key independent variable was the time care was received which was categorized as before the pandemic started, and after the pandemic started. Drawing from previous studies, potential confounding variables such as the respondents' age, marital status, the highest level of education, religion, number of previous pregnancies, number of living children, knowledge of COVID-19, place of residence, and State of residence, were included as control variables.

Difficulty in accessing care during the periods was another outcome variable which was accessed with multiple response questions on a set of difficulties consisting transportation, no money to pay for services, no drugs, no family planning products, family planning product of choice not available, availability of providers, and others. Each difficulty was recoded into a

dummy variable where 1 is an experience of a particular difficulty and 0 is otherwise. A binary index of experience of difficulty was generated by aggregating the scores: experience of difficulty was coded 1 whereas the experience of no difficulty was coded as 0. The independent variables included age, marital status, highest level of education, State of residence, place of residence, number of previous pregnancies, and number of living children.

## Data management and analysis

The data were extracted from the ODK to Statistical Package for Social Science (SPSS) version 20 for data cleaning and analysis. The data are available on Zenodo, an open repository 10.5281/zenodo.6128442. The descriptive results are presented as absolute numbers, percentage, mean and standard deviation, and range where appropriate. Using binary logistic regression, a further analysis was conducted to examine the probability of utilizing each of the RMNCH services before and after the pandemic started. Five models were estimated using each of the five RMNCH services as the outcome variable. Concerning the experience of difficulty, a distribution of experience of difficulty before and after the pandemic started was presented using absolute numbers and percentages. To ascertain if the experience of difficulties accessing RMNCH services varied significantly by the two periods and respondents' characteristics two binary logistic regression models were estimated. One model estimated the odds of experiencing difficulty in accessing RMNCH services after the pandemic started compared to before the pandemic and the experience of difficulty by the selected characteristics of the respondents for each of the two periods.

The personal characteristics of the respondents, State and place of residence were adjusted in logit models for utilization of each RMNCH service, and the comparison between the two periods in experience of difficulty. All the statistical analyses were two-tailed with a 95% confidence interval, and the alpha was set at 0.05. The responses to the open-ended questions were reported narratively.

## Ethical consideration

Ethical approvals to conduct the study were obtained from the Ministries of Health in each state, and in some states were complemented by permission sought from Advisers on Health to the State Governments. Also, the Research Ethics Committee of the College of Medical Sciences, University of Benin, Nigeria approved the study #CMS/REC/2020/092 dated October 4, 2020. Further approvals to conduct the study were obtained from the heads of the communities and the facilities visited, and only those who consented were included in the study. Finally, the study was fully explained to each respondent, and they were informed that the information they provide would only be used for the study and not for anything else. They were also assured of the confidentiality of information they provide and that their names will not feature in the report or any publications. Informed consent was a part of the questionnaire, and data were collected using computer-assisted personal interviewing, thus, verbal consent was obtained and ticked after the respondent had given consent. This method was approved by the various Ministries of Health. The research did not involve minors, thus, obtaining consent from parents or guardians was not applicable.

# Results

## Description of the clients by State and socio-demographic characteristics

Out of the 3840 expected respondents, 2930 were successfully interviewed, giving a response rate of 76.3%. The distribution of respondents by State and their socio-demographic

characteristics are presented in Table 1. As shown, the mean (SD) age of the respondents was 29.8 (8.5) years. The majority were married in a formal union (81.2%), whereas 14% were never married. Many attained secondary (42.5%) and tertiary education (34.1%). Slightly more than half were of the Islamic religion, while 44.1% were Christians. More than a quarter lived in rural areas (27.2%) whereas 41.5% lived in urban areas, and about one-third lived in semi-urban areas.

Slightly more than half of the respondents registered in the respective PHCs before the pandemic started. Slightly above one-fifth (22%) of the respondents were nulliparous, and a larger proportion has had 1–3 pregnancies. The mean number of pregnancies was 2.4±2.3, and the mean number of living children was 2.3±2.1.

## Reproductive, maternal, neonatal, and child health (RMNCH) services received in the PHCs

The pattern of utilization of PHCs for RMNCH services before and after the pandemic started is presented in Table 2. More than half of the respondents reported receiving family planning, antenatal care, delivery care, and child health services before the pandemic. The majority reported receiving postnatal care after the pandemic started.

Adjusting for the confounders, the odds of utilizing family planning services before the pandemic was 56% higher than after the pandemic started (Table 3). Antenatal, and delivery care services were significantly more likely to be utilized before than after the pandemic started. In contrast, the likelihood of utilizing postnatal care was 38% lower before than after the pandemic started.

## RMNCH services access, difficulties, missed appointments, and response before and after COVID-19 started

In Table 4, access, difficulties, missed appointments, and use of alternative services before and after the pandemic started are presented. The average number of visits to the PHCs before the pandemic was once a week, while the common challenges mentioned were no money to pay for services (user fees), transportation difficulties, and the lack of drugs. Other difficulties experienced before the pandemic included distances to PHCs, poor attitude of health care providers, long waiting times for service delivery, delay in getting test results, financial constraints, health workers not available, high price on drugs, lack of water in the facility, poor sanitary conditions, limited privacy, and lack of space in the clinic.

By contrast, after the pandemic started, 9.7% of the patients reported having problems attending the clinic. The frequency of visits was reduced to less than one per week, on average. The commonly mentioned difficulties reported by women in using the PHCs during the pandemic included providers not available (69.6%), no drugs (24.5%), and no family planning products (21.8%). Most missed appointments were for antenatal care (45.8%), family planning services (27.7%), and child immunization (23%). For those missing RMNCH appointments, 16.4% of the respondents used another facility, the majority (74.3%) did nothing, while others called providers known to them, rescheduled the appointments, or used nearby pharmacies.

In Table 5, a distribution of the experience of difficulty in accessing RMNCH services before and after the pandemic started is presented by the characteristics of the respondents; and the odds of experiencing difficulty by the different categories of women were presented in Table 6. Most of the respondents who experienced difficulty accessing RMNCH services were aged 20–39 in both periods. However, the odds of experiencing a difficulty were only statistically significant by age after the pandemic started. The respondents who were less than 19 years old, 25–29, and 30–34 years old were significantly less likely to report that they

**Table 1. Distribution of clients by State, socio-demographic, and obstetrics characteristics.**

| Characteristic | Frequency | Percent |
|---|---|---|
| | (n = 2930) | |
| **State** | | |
| Akwa Ibom | 242 | 8.3 |
| FCT | 322 | 11.0 |
| Borno | 61 | 2.1 |
| Enugu | 289 | 9.9 |
| Gombe | 448 | 15.3 |
| Kaduna | 323 | 11.0 |
| Kano | 262 | 8.9 |
| Lagos | 300 | 10.2 |
| Ogun | 251 | 8.6 |
| Sokoto | 432 | 14.7 |
| **Age** | | |
| Range 0–68 | | |
| Mean (sd) | 29.8(8.5) | |
| **Marital status** | | |
| Married | 2377 | 81.2 |
| Never married | 413 | 14.1 |
| Separated | 39 | 1.3 |
| Divorced | 41 | 1.4 |
| Living with a partner | 25 | 0.9 |
| Widowed | 32 | 1.1 |
| **Highest level of education** | | |
| Primary | 267 | 9.7 |
| Secondary | 1173 | 42.5 |
| Tertiary | 941 | 34.1 |
| Vocational | 151 | 5.5 |
| Islamic education | 225 | 8.2 |
| **Religion** | | |
| Christian | 1291 | 44.1 |
| Muslim | 1617 | 55.2 |
| Traditional | 19 | 0.6 |
| Others | 3 | 0.1 |
| **Place of residence** | | |
| Rural | 797 | 27.2 |
| Semi-urban | 916 | 31.3 |
| Urban | 1216 | 41.5 |
| **Time care was received** | | |
| Before covid-19 started | 1502 | 51.3 |
| After covid-19 started | 1427 | 48.7 |
| **Number of previous pregnancies** | | |
| 0 | 645 | 22.0 |
| 1 | 500 | 17.1 |
| 2 | 563 | 19.2 |
| 3 | 478 | 16.3 |
| 4 | 277 | 9.5 |
| 5 | 203 | 6.9 |

(*Continued*)

**Table 1.** (Continued)

| Characteristic | Frequency | Percent |
|---|---|---|
| | (n = 2930) | |
| 6–16 | 263 | 9.0 |
| Mean (SD) | 2.4(2.3) | |
| **Number of previous abortions** | | |
| 0 | 2631 | 89.8 |
| 1 | 195 | 6.7 |
| 2 | 66 | 2.3 |
| 3–8 | 37 | 1.2 |
| Mean (SD) | 0.2(0.5) | |
| **Number of living children** | | |
| 0 | 671 | 22.9 |
| 1 | 530 | 18.1 |
| 2 | 605 | 20.7 |
| 3 | 470 | 16.0 |
| 4 | 268 | 9.1 |
| 5 | 186 | 6.4 |
| 6–15 | 199 | 6.8 |
| Mean (SD) | 2.3(2.1) | |

Note: The frequency for some of the variables is slightly below 2930 due to omission in the dataset

experienced difficulty accessing RMNCH services after the pandemic started compared to older women aged 45–49 years. The majority (77% and above) of the respondents in the two periods were in a union (married and living together with a partner). Marital status was not a significant factor in the experience of difficulty. Before and after the pandemic started, reporting of difficulty accessing RMNCH services was more among respondents who had attained secondary and higher education, but the relationship was only significant for women who attained vocational education compared to Islamic education (OR 2.47 CI:1.35–4.31).

The distribution by States indicates that most of the respondents who reported difficulty accessing RMNCH services before the pandemic were from Gombe, Enugu, FCT, and Lagos. After the pandemic started, most of those who reported difficulty were from Enugu, Gombe, Akwa Ibom, and Sokoto. Compared to the FCT, the likelihood of experiencing difficulty accessing RMNCH services in the PHCs before the pandemic was significantly higher in Akwa Ibom, Enugu, and Gombe, and significantly lower in Ogun State. After the pandemic started, this pattern remained except for Ogun, respondents in Akwa Ibom, Enugu and Gombe

**Table 2. Utilization of RMNCH before and after the pandemic started (Multiple responses).**

| Service | Before the pandemic | After the pandemic started |
|---|---|---|
| | N (%) | N (%) |
| Family Planning | 618(56.6) | 474 (43.4) |
| Antenatal Care | 765(51.8) | 712 (48.2) |
| Delivery Care | 501(55.6) | 400 (44.4) |
| Postnatal Care | 147(37.5) | 245 (62.5) |
| Child Care | 686(52.1) | 630 (47.9) |
| Total | 2717(52.5) | 2461(47.5) |

**Table 3. The odds of utilizing RMNCH services before the pandemic versus after the pandemic started.**

| Service | Odds Ratio (95% CI) | p-value |
|---|---|---|
| Family Planning | 1.56(1.30–1.87) | <0.001 |
| Antenatal Care | 1.19(1.00–1.43) | 0.050 |
| Delivery Care | 1.52(1.26–1.84) | <0.001 |
| Postnatal care | 0.62(0.48–0.80) | <0.001 |
| Child Care | 0.96(0.81–1.14) | 0.667 |

Note: The omnibus goodness of fit test was highly significant in all the models.

**Table 4. RMNCH services access, difficulties, missed appointments, and response before and after COVID-19 started.**

| Variable | Frequency | Percent |
|---|---|---|
| **Frequency of visits to the PHC before the pandemic per week** | | |
| Mean (SD) | 1.02(0.75) | |
| **Difficulty in attending the clinic before the pandemic (Multiple responses)** | | |
| No transportation | 872 | 29.8 |
| No money to pay for services | 912 | 31.2 |
| No drugs | 417 | 14.2 |
| No FP product | 114 | 3.9 |
| FP product of choice not available | 190 | 6.5 |
| Other | 1237 | 42.3 |
| **Had a problem attending the clinic after covid-19 started** | | |
| Yes | 285 | 9.7 |
| **Difficulties in attending clinic after covid-19 started (multiple responses)** | | |
| Providers were not available | 483 | 69.6 |
| No drug | 170 | 24.5 |
| No FP product | 151 | 21.8 |
| Other | 82 | 11.8 |
| **Frequency of visits after pandemic started per week** | | |
| Range 0–21 | | |
| Mean (SD) | 0.9(0.8) | |
| **Missed appointment after the pandemic started (multiple responses)** | | |
| Family planning | 112 | 27.7 |
| Antenatal care | 185 | 45.8 |
| Delivery care | 28 | 6.9 |
| Postnatal care | 36 | 8.9 |
| Child immunization | 93 | 23.0 |
| Other | 37 | 9.2 |
| **Response about the missed appointment** | | |
| Used another facility | 23 | 16.4 |
| Did nothing | 104 | 74.3 |
| Used TBA | 5 | 3.6 |
| Other | 8 | 5.7 |

Note: Other difficulties include distance, the attitude of healthcare providers, crowd and slow service, husband's permission.

**Table 5. Distribution of experience of difficulties by the respondents' characteristics and State.**

| Characteristic | Reported difficulty before the pandemic | | Reported difficulty after the pandemic started | |
|---|---|---|---|---|
| | Yes | No | Yes | No |
| | N(%) | N(%) | N(%) | N(%) |
| **Experienced difficulty** | 307(20.4) | 1195(79.6) | 387(27.1) | 1040(72.9) |
| **Age** | | | | |
| <19 | 26(8.5) | 119(10.0) | 17(4.4) | 79(7.6) |
| 20–24 | 45(14.7) | 225(18.8) | 58(15.0) | 170(16.3) |
| 25–29 | 76(24.8) | 328(27.4) | 93(24.0) | 317(30.5) |
| 30–34 | 65(21.2) | 246(20.6) | 69(17.8) | 232(22.3) |
| 35–39 | 56(18.2) | 140(11.7) | 74(19.1) | 143(13.8) |
| 40–44 | 19(6.2) | 82(6.9) | 39(10.1) | 51(4.9) |
| 45–49 | 20(6.5) | 55(4.6) | 37(9.6) | 48(4.6) |
| **Marital status** | | | | |
| Not in a union | 42(13.7) | 267(22.4) | 53(13.7) | 163(15.7) |
| In a union | 265(86.3) | 926(77.6) | 334(86.3) | 876(84.3) |
| **Highest level of education** | | | | |
| Primary | 39(13.4) | 96(8.4) | 44(12.0) | 88(9.1) |
| Secondary | 107(36.9) | 484(42.5) | 156(42.6) | 425(44.2) |
| Tertiary | 110(37.9) | 418(36.7) | 91(24.9) | 322(33.5) |
| Vocational | 10(3.4) | 52(4.6) | 47(12.8) | 42(4.4) |
| Islamic education | 24(8.3) | 88(7.7) | 28(7.7) | 85(8.8) |
| **State** | | | | |
| Akwa Ibom | 35(11.4) | 54(4.5) | 73(18.9) | 80(7.7) |
| Borno | 17(5.5) | 38(3.2) | 1(0.3) | 5(0.5) |
| Enugu | 39(12.7) | 64(5.4) | 82(21.2) | 104(10.0) |
| Gombe | 68(22.1) | 162(13.6) | 75(19.4) | 143(13.8) |
| Kaduna | 21(6.8) | 148(12.4) | 37(9.6) | 117(11.2) |
| Kano | 27(8.8) | 202(16.9) | 7(1.8) | 26(2.5) |
| Lagos | 36(11.7) | 110(9.2) | 20(5.2) | 134(12.9) |
| Ogun | 8(2.6) | 152(12.7) | 10(2.6) | 81(7.8) |
| Sokoto | 18(5.9) | 89(7.4) | 58(15.0) | 266(25.6) |
| FCT | 38(12.4) | 176(14.7) | 24(6.2) | 84(8.1) |
| **Place of residence** | | | | |
| Rural | 98(31.9) | 215(18.0) | 181(46.8) | 303(29.1) |
| Semi-urban | 107(34.9) | 329(27.5) | 131(33.9) | 349(33.6) |
| Urban | 102(33.2) | 651(54.5) | 75(19.4) | 388(37.3) |
| | **Mean (Standard deviation)** | | | |
| **Number of previous pregnancies** | 2.57(2.29) | 2.16(2.28) | 2.87(2.21) | 2.55(2.22) |
| **Number of living children** | 2.49(2.21) | 1.98(2.11) | 2.77(2.01) | 2.32(1.96) |

remained significantly more likely to report that they experienced difficulty accessing RMNCH services in the PHCs. Most of those who reported experiencing any of the difficulties before the pandemic were semi-urban residents, although the difference between them, the rural and urban residents was not large. On the contrary, after the pandemic started, most of the respondents who reported experiencing difficulties were from rural areas. The odds of experiencing difficulties were significantly more likely among rural residents compared to urban residents both before the pandemic and after the pandemic started. The respondents

**Table 6. Logistic regression analysis of the experiencing difficulties in access RMNCH services before and after the pandemic started.**

| Variable | Before the pandemic | After the pandemic started |
|---|---|---|
| | OR(95% CI) | OR(95% CI) |
| **Age** | | |
| <19 | 0.61(0.25–1.48) | 0.38(0.16–0.92)* |
| 20–24 | 0.65(0.30–1.41) | 0.55(0.29–1.07) |
| 25–29 | 0.62(0.31–1.26) | 0.49(0.27–0.88)* |
| 30–34 | 0.58(0.29–1.16) | 0.47(0.26–0.85)* |
| 35–39 | 1.03(0.52–1.07) | 0.75(0.42–1.34) |
| 40–44 | 0.56(0.25–1.26) | 1.06(0.54–2.08) |
| 45–49 (Ref) | 1 | 1 |
| **Marital status** | | |
| Not in a union | 0.68(0.45–1.02) | 0.94(0.61–1.44) |
| In a union (Ref) | 1 | 1 |
| **Highest level of education** | | |
| Primary | 1.66(0.87–3.16) | 1.16(0.63–2.15) |
| Secondary | 1.15(0.65–2.03) | 0.96(0.57–1.61) |
| Tertiary | 1.48(0.81–2.70) | 0.98(0.56–1.71) |
| Vocational | 1.01(0.41–2.49) | 2.47(1.22–4.97)* |
| Islamic education (Ref) | 1 | 1 |
| **State** | | |
| Akwa Ibom | 2.39(1.31–4.36)** | 2.83(1.56–5.14)** |
| Borno | 1.72(0.74–4.01) | 3.04(0.24–37.9) |
| Enugu | 2.36(1.29–4.31)** | 2.41(1.35–4.31)** |
| Gombe | 2.01(1.21–3.34)** | 2.53(1.38–4.63)** |
| Kaduna | 0.64(0.35–1.16) | 1.37(0.73–2.57) |
| Kano | 0.84(0.47–1.49) | 1.92(0.68–5.43) |
| Lagos | 1.34(0.75–2.39) | 0.63(0.32–1.27) |
| Ogun | 0.29(0.13–0.65)** | 0.67(0.28–1.62) |
| Sokoto | 1.08(0.55–2.11) | 1.14(0.61–2.10) |
| FCT (Ref) | 1 | 1 |
| **Place of residence** | | |
| Rural | 2.04(1.36–3.06)** | 2.44(1.66–3.59)*** |
| Semi-urban | 1.21(0.80–1.82) | 1.52(1.02–2.28)* |
| Urban (Ref) | 1 | 1 |
| **Number of previous pregnancies** | 0.97(0.88–1.08) | 0.97(0.87–1.09) |
| **Number of living children** | 1.06(0.94–1.19) | 1.07(0.94–1.22) |

Note: OR–Odds Ratio; CI: Confidence Interval;

*p<0.05,

**p<0.01,

***p<0.001

who reported experiencing any of the difficulties accessing RMNCH services were women who have had an average of 3 pregnancies and children during the two periods. These had no significant relationship with the experience of difficulty accessing RMNCH services in the PHCs.

When other factors were held constant, the experience of difficulty accessing RMNCH services was 23% more likely after the pandemic started than before the pandemic started (OR 1.23, p<0.05 CI: 1.01–1.50).

## Recommendations from respondents

The respondents were asked what recommendations they would make to the government and other responsible agencies for the provision of family planning and other reproductive health services during the pandemic. The summary of the many responses is presented in Table 7. The commonly mentioned recommendations included the need to employ more health workers, and the establishment of more PHCs to reduce waiting times. Other recommendations were the provision of affordable fees or free services, removal of barriers on pregnant women and children seeking health facilities during the lockdown, and the training of law enforcement agents to understand the RMNCH needs of the populace. Also recommended, among others was the provision of more commercial buses to ease transportation difficulties and to ensure social distancing due to crowded transportation during the pandemic.

## Discussion

The study was designed to investigate the experiences and perceptions of women in seeking RMNCH services before and during the COVID-19 pandemic, and to draw lessons for the management of such services during the pandemic and others that may occur in the future. The results indicate that women were less likely to use RMNCH services during the pandemic as compared to the period before the pandemic. While the women had at least one clinic visit per week before the pandemic, this was reduced to less than 0.9 visits per week after the

**Table 7. Recommendations from respondents.**

| | | |
|---|---|---|
| Employ more health attendants/ providers<br>Build more PHCs | Law enforcement officers should temper justice with mercy | Affordable healthcare for infants and nursing mothers<br>Free maternal and child health services |
| Ensure that patients in public hospitals keep social distancing | A good approach to family planning is needed from the government especially in engaging traditional and religious leaders for robust family planning uptake | Allow pregnant women and children to go to hospital during lockdown without restrictions<br>Assist mothers financially |
| Conducting other tests aside from pregnancy tests before giving Family planning | Ensure availability of FP commodities and consumables<br>Family planning products should be distributed free. | Provide buses for the masses so that people will keep social distancing while in commercial buses |
| Provision of adequate and free drugs in the facility | Government should make everything work better in the facility to ensure safe delivery and maternal care | Need for more reorientation of the staff |
| Create more awareness about the COVID-19 disease and provide hygiene materials | Adequate facilities/equipment for the PHCs | Produce free nose masks for clinic days to save costs for pregnant mothers. |
| Government should build more PHC to make it closer to the people | Government should always provide the health center with FP and PPEs facilities to help them fight and secure the community from the corona virus. Again, the government should recruit professionals in the field, employ health workers, and pay them high so that they can put in their lives to protect and serve the community well as COVID-19. | Palliative should be provided for users of PHC to help close the nutritional gap. |

pandemic started. The multivariable regression analysis showed a 56%, 19%, and 52% increased likelihood of utilization before the pandemic of family planning, antenatal, and delivery services respectively, while there was a 38% reduced likelihood for postnatal services. Similar reduced odds of RMNCH utilization has been reported in past studies in other middle-low income countries, and vulnerable groups such as refugees [16–19]. The increased use of postnatal services despite the pandemic may be due to the relatively non-invasive nature of postnatal visits, and the tendency for it to be routine rather than being an emergency visit. The design of antenatal, delivery, child health, and postnatal services during the pandemic would need to be reconsidered, perhaps to include elements such as home visits or e-health components that address barriers and facilitate access to essential RMNCH services during the pandemic [19, 20].

The results show that some of the difficulties reported by women in using PHCs for RMNCH services during the pandemic included "providers not available" and the lack of family planning commodities. Although several reports have suggested that the pandemic might limit the ability of health facilities to provide RMNCH services [1, 2, 21–23], the results of this study provide substantive evidence from the perspectives of clients that health providers and products may not be available in health facilities due to the pandemic. It is evident that novel methods would need to be devised to increase the availability and responsiveness of health workers for RMNCH during such pandemics and to ensure the uninterrupted delivery of commodities during the period.

We further solicited information from the respondents on the alternative sources of services they sought in the face of the challenges they experienced in receiving services from the PHCs clinics during the period. The results showed that while a substantial proportion "did nothing", meaning they did not seek alternative sources of care, some called other service providers, while some re-scheduled their appointments or used nearby pharmacies. The proportion doing nothing is worrisome as it meant that women were exposed to the risk of unwanted pregnancies and other adverse reproductive health outcomes. The reported use of pharmacies by the women was of interest. The role of pharmacists in providing SRH services has been documented in a previous study in four Eastern Mediterranean countries [24]. Although pharmacies do not normally provide RMNCH services, this response indicates that they could be re-trained to provide such essential services at times of emergencies such as that posed by COVID-19.

A critical component of this study was the response to the question on recommendations for addressing RMNCH delivery during the pandemic. We conjectured that this was critically important to allow for bottom-to-top decision making on steps to be taken into consideration to consolidate the delivery of services during the pandemic. With such inputs from beneficiaries of RMNCH interventions, we argue that it would encourage the identification and uptake of the most relevant and appropriate policies and programs aimed at sustaining RMNCH delivery during the pandemic. The results of the analysis of the recommendations made by the respondents can be summarized under three broad categories, as follows: 1) the need for facility improvement, and staff recruitment and re-training; 2) free and readily available PHC services including family planning, maternal and child health services, and 3) provision of social safety nets including transportation and palliatives. Respondents decried the limited PHCs in their immediate communities as too few to cater for women requiring RMNCH services. This was compounded by the limited number of health personnel in the few available facilities, with many health workers often not present on a 24-hour basis or not fully motivated to work. Part of the problem is the rural location of the facilities where health workers may not have the necessary incentives to work, or as we earlier reported in a review of COVID-19 response in PHCs [8], or the fact that many of the facilities do not have personal preventive equipment

(PPEs) to prevent being infected with COVID-19. Thus, a major requirement in the response to COVID-19 response in PHCs is for the government to invest in health workers' recruitment, and re-training, and the provision of incentives and PPEs to motivation, competency, and agency to address RMNCH services during the pandemic.

Several respondents made recommendations on the need for more PHC facilities that are accessible and affordable, to promote the use of services by women. Many recommended that PHC services, especially family planning and COVID-19 services should be free to citizens and paid for by the government. Given the high rate of poverty in Nigeria's rural population [25], the almost absent health insurance policies, and the heavy reliance on out-of-pocket health payments in the country [26, 27], we posit that this recommendation should be seriously considered by the government and relevant stakeholders. This is more critical in the face of the severe economic hardships associated with the pandemic [28, 29], and the fact that many households are unable to meet their everyday socio-economic requirements.

This would align with the third group of recommendations made by the respondents for improving the delivery of RMNCH services during the COVID-19 pandemic, which is the provision of social safety nets including transportation and palliatives to women and families. While this has featured as a national response to the pandemic in Nigeria [30, 31], there is a need to expand its scope to reach broader categories of women especially those in rural areas. More importantly, it would be critical and more effective to link such palliatives to conditional transfers to leverage additional benefits and women's responsiveness in the use of PHC services. As an example, the provision of transportation during the pandemic will greatly help women overcome some of the barriers and difficulties they experience in accessing services during the pandemic.

## Strengths and weaknesses

To the best of our knowledge, this is one of a few studies that have investigated women's perceptions and experiences on the use of RMNCH services during the COVID-19 pandemic. The focus on PHC services in rural communities is novel as it enables the elucidation of services in the first level frontline health care system, which has wider accessibility for the majority of vulnerable women and children. The study of women in rural communities places the issues of equity and social vulnerability on the front burner and the importance to reach all categories of women and children as a strategy for attaining universal health coverage. The study coverage of health facilities in the six geo-political zones of the country also allows for the external generalizability of the results for the development and implementation of policies and programs on a countrywide basis.

Despite the strong characteristics, the study is limited by its use of only the questionnaire survey methodology, rather than a mixed design that includes qualitative research. The inclusion of qualitative research through focus groups discussion, and perhaps community conversation would have provided greater depths and meanings to the challenges faced by women in accessing RMNCH services during the pandemic and in elucidating the specific recommendations and perspectives of women. Nevertheless, the results of this study allow a rich mix of lessons from different categories of women across the country and would galvanize the wider use of the results for the design and implementation of nationwide policies and programs for improving women's access to RMNCH services during COVID-19 pandemic.

## Conclusion

We conclude that the COVID-19 pandemic limits women's access to RMNCH antenatal, delivery, and childcare services offered in PHCs in nine states and FCT drawn from the six

geopolitical zones of Nigeria. Women perceived that the improvement and expansion of PHC services, the provision of social safety nets, free services, and palliatives would limit the impact of the pandemic on the use of RMNCH services for vulnerable women and children. We believe that addressing these concerns will help to sustain the delivery of RMNCH services during the COVID-19 pandemic.

## Supporting information

**S1 File.**
(DOCX)

## Acknowledgments

We wish to thank the commissioner and officials of the state ministries of health and States Primary Health Care Development Agencies in Akwa Ibom, Borno, Enugu, Gombe, Lagos, Kaduna, Kano, Ogun, Sokoto, FCT, and Akwa Ibom State Department of Multilateral and Donor Agency for their support and for granting access to the health facilities. We thank Dr Dorothy Ononokpono, Joy Nee, and Dr Christopher Okafor who supervised, collated, and ensured data quality in the 10 states and the data collectors from the various states. We are grateful to the all the Civil Society Organizations; Community Health and Research Initiative (CHR), Women widows and Orphans Development Initiative, Youth Future Saver Initiative, Positive Outreach Foundation Contact, Helping Hands and Grassroot Support Foundation, Mariya Tambuwal Development Initiative, Hope Initiative for Vulnerable and Marginalised, Chabash Development and Health Initiative, Hacey Health Initiative, Star Girl Education Foundation, Applicant Welfare and Development Centre, Family Empowerment & Youth Reorientation Path Initiative, Global Health Awareness Research Foundation, Royal Health-care Foundation, New Life Community Care Initiative, Saif Advocacy Foundation, Minds for Environment, Health and Community Initiative, Wildan Care Foundation, Galaxy Women, Advocacy Center for Development, Center for Grassroots Community Education and Development, New Age Initiative for Youth Development, Life and Hope Initiative, Care For Life Mission, Voice of the Girl Child and Vulnerable People Foundation, Women and Children's Right and Empowerment Foundation, Hope for Communities and Children Initiative, Arewa Girls Initiative, Deborah Women and Youths Foundation, GAIL Foundation, Global initiative for Women and Children, Association of Positive Youth Living with HIV in Nigeria KAD-APYIN, Eagle Lead Development Initiative, Bako Youth Development Foundation, Strong Enough Girls Empowerment Initiative, Abiodun Essiet Initiatives for Girls, Dorothy Njemanze Foundation, Lola Cater for the needy foundation, Crestville Development Foundation who served as community focal persons in all the project LGAs. They were instrumental in helping the project teams to gain access to the project communities.

## Author Contributions

**Conceptualization:** Babatunde Adelekan, Erika Goldson, Osaretin Adonri, Yakubu Aliyu, Matthew Onoja, Idowu Araoyinbo, Emilene Anakhuekha, Ulla Mueller, Friday E. Okonofua.

**Data curation:** Lorretta Favour C. Ntoimo, Friday E. Okonofua.

**Formal analysis:** Lorretta Favour C. Ntoimo, Friday E. Okonofua.

**Funding acquisition:** Babatunde Adelekan, Erika Goldson, Osaretin Adonri, Yakubu Aliyu, Matthew Onoja, Idowu Araoyinbo, Emilene Anakhuekha, Ulla Mueller.

**Investigation:** Lorretta Favour C. Ntoimo, Eno-Obong Ekwere, Micheal Inedu, Olayinka Moruf, George Swomen, Brian Igboin, Friday E. Okonofua.

**Methodology:** Lorretta Favour C. Ntoimo, Eno-Obong Ekwere, Micheal Inedu, Olayinka Moruf, George Swomen, Brian Igboin, Friday E. Okonofua.

**Project administration:** Babatunde Adelekan, Erika Goldson, Osaretin Adonri, Yakubu Aliyu, Matthew Onoja, Idowu Araoyinbo, Emilene Anakhuekha, Ulla Mueller, Eno-Obong Ekwere, Micheal Inedu, Olayinka Moruf, George Swomen, Brian Igboin, Friday E. Okonofua.

**Supervision:** Lorretta Favour C. Ntoimo, Eno-Obong Ekwere, Micheal Inedu, Friday E. Okonofua.

**Writing – original draft:** Lorretta Favour C. Ntoimo, Friday E. Okonofua.

**Writing – review & editing:** Babatunde Adelekan, Erika Goldson, Lorretta Favour C. Ntoimo, Osaretin Adonri, Yakubu Aliyu, Matthew Onoja, Idowu Araoyinbo, Emilene Anakhuekha, Ulla Mueller, Eno-Obong Ekwere, Micheal Inedu, Olayinka Moruf, George Swomen, Brian Igboin, Friday E. Okonofua.

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
