## [Decision Letter · Decision Letter 0]

27 Mar 2023

PONE-D-22-19181Clients’ perspectives on the utilization  of reproductive, maternal, neonatal, and child health services in primary health centers during COVID-19 pandemic in 10 States of Nigeria: A cross-sectional studyPLOS ONE

Dear Dr. Okonofua,

Thank you for submitting your manuscript to PLOS ONE. After careful consideration, we feel that it has merit but does not fully meet PLOS ONE’s publication criteria as it currently stands. Therefore, we invite you to submit a revised version of the manuscript that addresses the points raised during the review process.

We look forward to receiving your revised manuscript.

Kind regards,

Kehinde S. Okunade

Academic Editor

PLOS ONE

Journal Requirements:

2. Please provide additional details regarding participant consent. In the ethics statement in the Methods and online submission information, please ensure that you have specified whether: 1) whether the ethics committee approved the verbal/oral consent procedure, 2) why written consent could not be obtained, and 3) how verbal/oral consent was recorded.

Reviewers' comments:

Reviewer's Responses to Questions

**Comments to the Author**

1. Is the manuscript technically sound, and do the data support the conclusions?

Reviewer #1: Partly

2. Has the statistical analysis been performed appropriately and rigorously? 

Reviewer #1: I Don't Know

3. Have the authors made all data underlying the findings in their manuscript fully available?

Reviewer #1: Yes

4. Is the manuscript presented in an intelligible fashion and written in standard English?

Reviewer #1: Yes

5. Review Comments to the Author

Reviewer #1: -What was the basis or criteria of selection of the 10 communities in each LGA used as study sites?

-Was ethical approval for the study obtained from any Health Research and Ethics Committee (HREC)? Ethical approval is different from permission from health ministries and centers to perform a study. State clearly the HREC that approved the work with the approval number.

As regards the inclusion of women who registered as having received clinical services in the clinics per state before and after the pandemic started. It is important to state the range of years before the pandemic that was targeted in this study? Were women who had accessed these services 10years ago also identified and recruited?

-Table 2 claims to describe “the pattern of utilization of RMNCH services pre- and post-covid”. How did you make up for/exclude women who did not need some of these services before or after the pandemic? It is important to differentiate women who did not require the service from those who needed it but couldn’t access it. Reasons why some women in the study may not have needed antenatal care before the pandemic could be because they were not pregnant before the pandemic. A woman who utilized post-natal services before the pandemic may not have needed it after the pandemic because she had no delivery after the pandemic. How did you ascertain that “no-need for a service” has not influenced the pattern depicted in Table 2?

-Clearly state what confounders were adjusted for in Table 3?

- A copy of the study questionnaire should be included as a supplementary file to the reviewer to help assess the work better.

6. PLOS authors have the option to publish the peer review history of their article (what does this mean?). If published, this will include your full peer review and any attached files.

Reviewer #1: No

---

## [Author Response · Author response to Decision Letter 0]

10 May 2023

May 10, 2023

Dear Editor,

Thank you for considering our manuscript PONE-D-22-19181 titled “Clients’ perspectives on the utilization of reproductive, maternal, neonatal, and child health services in primary health centers during COVID-19 pandemic in 10 States of Nigeria: A cross-sectional study” for publication.

Please, see below our point-by-point response to the editor’s and reviewers’ comments. 

Sincerely,

Professor Friday E. Okonofua

Response to Editor’s comments

Journal Requirements

Response: The PLOS ONE style has been followed.

2. Please provide additional details regarding participant consent. In the ethics statement in the Methods and online submission information, please ensure that you have specified whether: 1) whether the ethics committee approved the verbal/oral consent procedure, 2) why written consent could not be obtained, and 3) how verbal/oral consent was recorded.

Response: Verbal consent was obtained since the data collection was by computer-assisted personal interviewing, and a statement of information and consent was a part of the questionnaire. This has been explained in the ethical approval section. 

Response to Reviewer’s Comments 

Reviewer #1: -What was the basis or criteria of selection of the 10 communities in each LGA used as study sites?

Response: The ten communities were selected purposively based on location of a PHC in the community.

-Was ethical approval for the study obtained from any Health Research and Ethics Committee (HREC)? Ethical approval is different from permission from health ministries and centers to perform a study. State clearly the HREC that approved the work with the approval number.

Response: Approval to conduct this study was obtained from the various Ministries of Health in each State, and from the Research Ethics Committee, College of Medical Sciences, University of Benin, Nigeria #CMS/REC/2020/092 dated October 4, 2020.

As regards the inclusion of women who registered as having received clinical services in the clinics per state before and after the pandemic started. It is important to state the range of years before the pandemic that was targeted in this study? Were women who had accessed these services 10years ago also identified and recruited?

Response: There are three selection criteria. The second one is women receiving RMNCH and family planning services in the PHCs indicating that they were also using these services at the time of interview. 

-Table 2 claims to describe “the pattern of utilization of RMNCH services pre- and post-covid”. How did you make up for/exclude women who did not need some of these services before or after the pandemic? It is important to differentiate women who did not require the service from those who needed it but couldn’t access it. Reasons why some women in the study may not have needed antenatal care before the pandemic could be because they were not pregnant before the pandemic. A woman who utilized post-natal services before the pandemic may not have needed it after the pandemic because she had no delivery after the pandemic. How did you ascertain that “no-need for a service” has not influenced the pattern depicted in Table 2?

Response: The study population were women who were using a PHC in their community for any of RMNCH services. It excludes women who had no need of the RMNCH services. 

-Clearly state what confounders were adjusted for in Table 3?

Response: The confounders adjusted in the regression presented in Table 3 were listed in the methods section under variables and measures. They are respondents’ age, marital status, the highest level of education, religion, number of previous pregnancies, number of living children, knowledge of COVID-19, place of residence, and State of residence. 

- A copy of the study questionnaire should be included as a supplementary file to the reviewer to help assess the work better.

Response: A copy of the questionnaire has been included as a supplementary material.

---

## [Decision Letter · Decision Letter 1]

5 Jul 2023

Clients’ perspectives on the utilization  of reproductive, maternal, neonatal, and child health services in primary health centers during COVID-19 pandemic in 10 States of Nigeria: A cross-sectional study

PONE-D-22-19181R1

Dear Dr. Okonofua,

We’re pleased to inform you that your manuscript has been judged scientifically suitable for publication and will be formally accepted for publication once it meets all outstanding technical requirements.

Kind regards,

Kehinde S. Okunade

Academic Editor

PLOS ONE
---

## [Editor Report · Acceptance letter]

10 Jul 2023

PONE-D-22-19181R1 

Clients’ perspectives on the utilization of reproductive, maternal, neonatal, and child health services in primary health centers during COVID-19 pandemic in 10 States of Nigeria: A cross-sectional study 

Dear Dr. Okonofua:

I'm pleased to inform you that your manuscript has been deemed suitable for publication in PLOS ONE. Congratulations! Your manuscript is now with our production department. 

Kind regards, 

on behalf of

Dr. Kehinde S. Okunade 

Academic Editor

PLOS ONE